# Association of Quality of Coronal Filling with the Outcome of Endodontic Treatment: A Follow-up Study

**DOI:** 10.3390/dj5010005

**Published:** 2017-01-11

**Authors:** Manal Maslamani, Mai Khalaf, Amal K. Mitra

**Affiliations:** 1Department of Restorative Sciences, Faculty of Dentistry, Kuwait University, Safat 13110, Kuwait; 2Department of General Dental Practice, Faculty of Dentistry, Kuwait University, Safat 13110, Kuwait; mai.k@hsc.edu.kw; 3Department of Epidemiology & Biostatistics, School of Public Health, Jackson State University, Jackson, MS 39217, USA; amal.k.mitra@jsums.edu

**Keywords:** apical periodontitis, coronal filling, root filling, periapical index

## Abstract

The aim of this study was to assess the effect of the quality of the coronal restoration and the root filling on the success of endodontic treatment. Patients treated at Kuwait University Dental Clinics (KUDC) from 2003 to 2014 were contacted by telephone calls. Demographic data and clinical records of patients, including age, gender, the tooth number, and medical history were recorded. Each patient received clinical examination for all teeth, including assessment of the coronal filling (type, quality), root- and/or coronal fracture, and the periodontal condition around the tooth (e.g., probing depth, gingival recession); percussion and mobility tests. A periapical radiograph of the endodontic treated tooth was taken to determine the health of the periapical tissues using the periapical index. The quality of the root filling was assessed by length and density of the root filling. The mean follow-up period was 4.8 years. The overall success rate of endodontic treatment was 86%; teeth without any initial periapical lesion had a success of 93%, whereas those with such lesion had a success rate of 80%. Periapical healing was not significantly associated with either the length of root filling (*p* = 0.40) or the density of root filling (*p* = 0.099), but was statistically significantly associated with the presence of coronal filling defects (*p* = 0.001). This study demonstrated that inadequate coronal filling but not the quality of root filling was associated with a higher prevalence of periapical lesions.

## 1. Introduction

The outcome of endodontic treatment depends on the successful preparation of the root canal space by the removal of any tissue debris, complete destruction of all microorganisms, and a complete seal of the coronal and the apical ends of the root canal system [1,2]. Clinical assessment of both root canal treatment and coronal restoration is fundamental when evaluating endodontic treatment.

Current literature shows that the apical extent of root fillings should extend to within 0 to 2 mm of the radiographic apex. Much research has been done to compare short (>2 mm) and long filling (extruded beyond apex) to flush root fillings. A meta-analysis [3] indicated that flush fillings do exhibit the highest success rates, which minimize the likelihood of a foreign body reaction or microbial proliferation, hence affecting the outcome of endodontic treatment. The quality of a root filling affects the endodontic prognosis. Those radiographs which showed the presence of voids (a non-homogenous root filling) had significantly lower success rates than those without any radiographic indications of such voids [4]. There is unfortunately no standardization for this sort of measurement; however, significantly lower success rates have been demonstrated among those which were radiographically unsatisfactory [4]. In a Brazilian cohort study [5], poor root canal filling was found to be a prognostic determinant of endodontic treatment failure.

There is still some controversy when it comes to the level of impact of the coronal restoration on the success rate of the endodontic treatment. Much is known about the effect of permanent coronal restoration on the treatment outcome. A clinical study with known diagnostic and treatment factors [6] observed a tendency toward a higher success rate in teeth restored with permanent restoration, although the data were not statistically significant. Another study [7] observed that teeth that were permanently restored after orthograde endodontic or surgical retreatment had a significantly higher rate of successful outcome compared to the teeth that were not restored.

A study by Hommez et al. [8] followed the technique of Ray and Trope [9]; however, the former included a clinical evaluation of the coronal seal, and concluded that the odds of success comparing the quality of the root filling and coronal restoration were similar in magnitudes. The meta-analysis mentioned earlier [3] confirmed that the odds of success were significantly higher for restoration; these odds were found to be satisfactory compared with teeth with unsatisfactory restorations. The quality of coronal restoration postoperatively was a significant factor affecting the outcome of endodontics [3].

On the other hand, the quality of the coronal restoration (scored clinically and radiographically) did not have a significant influence on the periradicular status when it was combined with the quality of endodontic treatment [8]. Moreover, exposure of root fillings to the oral microbiota was not significantly correlated with periradicular status [10]. Also, the quality of the coronal restoration or the placement of post retentions did not affect the treatment outcome [10].

On the basis of the currently available evidence, the odds for healing of apical periodontitis increase with both adequate root canal treatment and adequate restorative treatment. Clinical outcomes may vary with a combination of adequate root filling and inadequate coronal restoration or a combination of inadequate root filling and adequate coronal restoration; however, no significant difference in the odds of healing was observed between these two combinations [11].

The purpose of the present study was to assess the prevalence of apical periodontitis in root canal-treated teeth after 1–11 years, and to evaluate clinically and radiographically the impact of the quality of root fillings and coronal restorations on periapical health.

## 2. Materials and Methods

This is a follow-up study conducted at the Kuwait University Dental Clinic (KUDC). From the clinical records of 350 patients treated by undergraduate students at KUDC from 2003 to 2014, everybody was approached by repeated telephone calls in 2014. Only 96 patients who agreed to participate after explaining the purpose of the study were enrolled. A written informed consent was taken before enrollment of each patient. All of these 96 patients came for a follow-up visit. The follow-up period was recorded for each individual from the time he/she underwent root filling up to the time of a follow-up examination in 2014.

The patients were seen by two examiners independently. Before the recall of patients for follow-up, variability between the examiners was examined for radiographic interpretation using the periapical index (PAI) of 10 periapical radiographs. Intra- and inter-examiner reliability was then assessed using Cohen’s Kappa statistics. In case of any disagreement, the two examiners discussed the case together and agreed on the clinical and radiographic findings.

At the follow-up examination, the clinical and radiographic findings were recorded by the same examiners using a structured form for each tooth. The radiographs were scored according to the PAI system. A scoring system of PAI was used based on an ordinal scale of five scores ranging from 1 being healthy to 5 having severe periodontitis with exacerbating features [12]. The scores were as follows: 1—normal periapical tissues, no bone changes; 2—small changes in bone structure; 3—changes in bone structure with some mineral loss; 4—periodontitis with well-defined radiolucent area; and 5—severe periodontitis with exacerbating features. The scores were dichotomized for each tooth to reflect absence (scores 1 and 2) or presence (scores 3–5) of apical periodontitis. Multi-rooted teeth were each given one score, which was the highest score for any of the roots.

Data including age, gender, the tooth number, number of teeth treated and medical history were recorded. Each patient was asked if he/she had any symptoms arising from the tooth, and was examined for all the teeth including the tooth which received root canal treatment. Clinical examination of the root canal treated tooth included assessment of the coronal filling (type, quality) using the modified index of [13], assessment for the presence of any root- and/or coronal fracture and the periodontal condition around the tooth (e.g., probing depth, gingival recession). The tooth was examined by tapping (percussion) to assess the health of the periapical tissues as well as the mobility of the tooth. The radiographic examination was performed using periapical radiographs. All radiographs were taken by the same radiographic technician using PM 2002 Proline radiographic equipment (Planmeca, Helsinki, Finland), and using the parallel technique.

The radiograph at the recall appointment was compared by the previous radiograph taken at the end of the treatment appointment to determine the health of the periapical tissues i.e., healing of previous periapical lesion, development of new periapical lesion, or the persistence of the same size of the periapical lesion. The health of the periapical tissue in the two radiographs was evaluated by the two examiners independently using PAI. Based on the periapical radiograph, the quality of the root filling was assessed also by the same specialists in terms of length, taper, and density of the root filling using an index. Correlation between the quality of the root filling and the presence or absence of the periapical lesion was evaluated.

The endodontic treatment was carried out by undergraduate students who were closely supervised by qualified endodontists. The ratio of undergraduate students to a qualified endodontist ratio was 3:1. The endodontic treatment protocol at KUDC was standardized for all the cases: (a) all treatments were under rubber dam; (b) teeth were isolated before the access cavity, using 2.5% sodium hypochlorite irrigation; (c) hand instrumentation was done using stainless steel files with the step-back technique and the obturation with lateral condensation technique using Sealapex sealer; and (d) the treatments were done in multiple visits. The coronal restoration was advised within one week after the root canal filling was performed.

### Statistical Analysis

Data were entered and analyzed using SPSS (version 23.0, Chicago, IL, USA). Descriptive analysis was done for demographic data. The relationship of length and density of the root filling, and the type and quality of the coronal filling were compared between the healthy and the diseased categories of PAI at follow-up by using Chi-square test. Fisher’s Exact test was used when the expected frequencies of the cells were small. A probability value of 0.05 or less was considered statistically significant.

## 3. Results

The Kappa score for inter-examiner agreement after the first calibration session was *k =* 0.8. After the second session performed one week later, the Kappa score for intra-examiner agreement was *k =* 0.9. Both scores indicated “good agreement”.

Information was available for 212 teeth of 96 patients. The mean ± SD age of the study patients was 39.6 ± 13.4 years, ranging from 15 to 69 years. Of them, 53% (50/95) were females. Of the 212 teeth examined, 13 (6%) were extracted, and six had poor radiographs which were inadequate for interpretation. Among 193 with available data, 91 (47%) had no lesions at baseline, while 100 (52%) had apical lesions, and only two (1%) had lateral lesions.

The follow-up period ranged from 1 to 11 years, with a mean ± SD of 4.8 ± 1.4 years. Table 1 shows the distribution of the number of patients and the length of the follow-up period according to the number of teeth treated. About 70% of patients were treated for either one or two teeth.

### 3.1. Periapical Index (PAI) and Success Rate

Table 2 shows that at baseline examination, 93/193 (48%) teeth were either normal (score 1) or they had small changes in bone structure (score 2). Out of the 193 teeth, 100 (52%) had a PAI score of 3–5, meaning that they were diseased. Of these 100 that were diseased, 64 had bone structure changes with small loss of minerals (2 mm) (score 3); 25 had periodontitis with well-visible radiolucent areas >2 mm (score 4); and 11 had an advanced form of periodontitis with exacerbating appearance >5 mm (score 5). At follow-up, the majority (166/193, 86%) were normal (score 1 and 2), which is considered the overall success rate. Among those who were recorded normal at baseline, the success rate was 93% (86/93) at follow-up, which was significantly higher than among those who had any PAI lesions and a success rate of 80% (80/100) (*p* = 0.006) (Table 2).

### 3.2. Relation between Length of Root Filling with PAI at Follow-up 

As shown in Table 3, out of 193 teeth examined, 141 (73%) had adequate length of root filling. No significant difference in the frequency of healthy teeth at follow-up was observed for the different lengths of root filling (Table 3).

In Table 4, it is shown that out of 193 teeth, 157 (81%) had adequate density (i.e., no voids). The frequency of healthy teeth at follow-up with adequate density of root fillings at baseline was 139/157 (89%), whereas the frequency of healthy teeth at follow-up with presence of void at baseline was 27/36 (77%). The difference was not statistically significantly (*p* = 0.099).

### 3.3. Relation between Coronal Filling with PAI at Follow-up

The majority type of coronal restoration was crown (121/193 (63%)), followed by composite or amalgam (56/191 (29%)) (Table 5). The frequency of healthy teeth at follow-up was 51/56 (91%) for teeth with composite or amalgam filling, 105/121 (85%) with crown, and 8/8 (100%) with temporary restoration. The teeth categorized as healthy or diseased at follow-up did not differ significantly in relation to the type of coronal filling (*p* = 0.322).

Regarding the presence of defects in the coronal fillings, 164/193, (85%) had no defects in the coronal filling. Among the teeth with no defects in coronal filling at baseline, 147/164 (90%) were healthy and only 17/164 (10%) were diseased at follow-up. The proportion of diseased teeth at follow-up with recurrent caries at baseline (7/22, 32%) was significantly higher than the diseased teeth without any defects at baseline (17/164, 10%) (Table 6).

## 4. Discussion

In this study, the quality of coronal restoration was significantly associated with the frequency of apical periodontitis at follow-up, but the type of restoration was not. There were significant differences in the frequency of healthy teeth (PAI 1 + 2) at follow-up for teeth with adequate restorations compared to non-adequate restorations or teeth with caries. However, there were no significant differences in the frequency of healthy teeth at follow-up for the different types of restoration. This is an interesting and important finding, especially since this study undertook both clinical and radiographic examinations and had a long follow-up period (1–11 years).

The overall success rate of normal PAI at follow-up was 86%, which is within the range of the reported success rate of the root canal treatment in other studies [2,3,13,14,15,16,17,18]. This success rate was higher than that (66%) reported by Tronstad et al. [13]. Obviously, the success rate was much higher among those who did not show any lesions at baseline compared to those that did (93% and 80%, respectively, *p* = 0.006).

The quality of the endodontic treatment performed at KUDC was generally of adequate length (125/167, 73%), and of adequate density (139/166, 82%). Hommes et al. 2002 found that 42% of the root canals were filled to an acceptable length, and 27% of them were performed by an apical periodontitis. Of the 58% of teeth which were not filled to an adequate length, 36% had apical periodontitis.

In our study, the teeth were assessed clinically and radiographically, in contrast to other studies which used radiographic assessment only [2,9,10,13,19,20,21,22,23,24], except one recent study reported by Dawson et al. (2016) who used both a clinical and a radiological assessment [25]. This raises the question about the accuracy of examining the quality of the coronal filling in earlier studies.

A study in a Danish population [19] found that teeth with an adequate root filling length were associated with a lower prevalence of periapical lesion than teeth with an inadequate length of the root filling (42.0% vs. 67.6%, respectively). Similarly, adequate coronal restorations were associated with lower prevalence of periapical lesion than inadequate restorations (48.0% vs. 63.9%, respectively). In the same study [19], the coronal restoration was assessed radiographically only. Results from the present study indicate that the periapical status of endodontically treated teeth depends on both the quality of the endodontic treatment and of the coronal restoration, and efforts should be made to optimize the treatment of teeth with pulpal or periapical infection. In a ten-year follow-up study, Kirkevang et al. [26] found that the root filling quality primarily affected the risk of persistent apical periodontitis. In a cross-sectional study in the Spanish population done by radiographic study only [20], 34.4% of the root fillings were adequate from a technical perspective. When both root fillings and coronal restorations were adequate, the prevalence of PAI decreased, as observed in the Korean population [24]. The quality of endodontic treatment and coronal restorations, classified by both radiographic and clinical evaluation, found that only approximately 36% of the teeth had adequate endodontic treatments and 68% had adequate coronal restorations [24]. Teeth with both adequate root fillings and restorations showed a significantly better outcome (82%), and teeth with both inadequate root fillings and restorations showed a significantly worse outcome (41%) than others [24]. The quality of the root filling, assessed radiologically in the Turkish population [27], found that 88% of the teeth were classified as having an inadequate root filling. This rate is much higher than what we found in our study. However, Da Silva et al. [28] did not find any significant association between the quality of the root filling and the presence of the periapical lesion in the Australian population.

There were several strengths and weaknesses of our study. Taking into consideration the small sample size and the fact that most cases were of adequate quality of root fillings, further studies with a larger sample size are needed. In this study, all subjects who were called by phone participated in the study, with a response rate of 100%. It is possibly because the treatments were offered free of cost. It is also worth noting that in the treatment protocol of this study, the permanent restoration was placed within one week of the root canal filling, as is recommended to reduce the chances for coronal leakage from the temporary filling [29]. In one in vitro study [30], bacterial products were found at the apex of the teeth after 3 weeks due to coronal leakage of root filled teeth without adequate coronal restorations.

Crowning of the endodontically treated tooth has been recommended in the literature for the protection of the remaining tooth structure [31]. This explains the high percentage (63%) of crown on the root canal treated teeth in this study. Although, in our study, the type of the coronal filling had no significant effect on the healing of the periapical lesion, the defect of the coronal filling had a significant effect on the healing. For example, 32% of the teeth with recurrent caries were diseased at follow-up. This figure was significantly higher than those that were diseased at follow-up but without any defects in coronal filling (10%). There is a strong consensus that caries or defective restorations in root-filled teeth allow leakage of oral microorganisms through the root canal to the periapical area where it causes inflammation [32]. This emphasizes the importance of coronal leakage as a cause of failure of root canal treatment.

## 5. Conclusions

The results of this study confirm that adequate coronal filling, assessed by both clinical and radiographic methods, had a greater impact on the postoperative periapical status, whereas the quality of root filling had a non-significant impact on the endodontic treatment outcome. A further controlled study with a larger sample size is suggested to compare the clinical outcome of the two groups of patients: one with the presence of an adequate root filling and an inadequate coronal restoration, and another with an inadequate root filling but an adequate coronal filling.

## Figures and Tables

**Table 1 dentistry-05-00005-t001:** Distribution of the number of patients and the length of the follow-up period according to the number of teeth treated.

No. Teeth Treated	Length of Follow-up Period (Year)		
	No. Patient (%)	Mean ± SD	Range
1 tooth	36 (37.5)	4.9 ± 1.3	4–9
2 teeth	31 (32.3)	5.3 ± 1.4	1–11
3 teeth	16 (16.7)	4.7 ± 1.0	2–7
4 teeth	7 (7.3)	4.9 ± 1.1	2–7
5 teeth	2 (2.1)	4.1 ± 0.7	3–5
6 teeth	2 (2.1)	4.5 ± 0.5	4–5
7 teeth	1 (1.0)	5.0	-
8 teeth	0 (0)	0	-
9 teeth	1 (1.0)	1.0	-
Total	96 (100.0)	4.8 ± 1.4	1–11

**Table 2 dentistry-05-00005-t002:** Periapical lesion at baseline and at follow-up.

	Periapical Lesion at Follow-Up		
Periapical Lesion at Baseline	Healthy (Score 1 + 2)	Diseased (Score 3−5)	Total
Healthy (Score 1 + 2)	86 (92%)	7 (8%)	93 (100.0%)
Diseased (Score 3 − 5)	80 (80%)	20 (20%)	100 (100.0%)
Total	166 (86%)	27 (14%)	193 (100.0%)

Chi-square test *p* = 0.006.

**Table 3 dentistry-05-00005-t003:** Relationship between the length of root filling and the periapical lesion (PAI) at follow-up.

	Periapical Lesion at Follow-Up		
Length of Root Filling	Healthy (Score 1 + 2)	Diseased (Score 3−5)	Total
1–2 mm	124 (88%)	17 (12%)	141 (100.0%)
>2 mm or no root filling	16 (76%)	5 (24%)	21 (100.0%)
Root filling material extruded	8 (80%)	2 (20%)	10 (100.0%)
At radiographic apex	18 (86%)	3 (14%)	21 (100.0%)
Total	166 (86%)	27 (14%)	193 (100.0%)

Fisher’s Exact test *p* = 0.403.

**Table 4 dentistry-05-00005-t004:** Relationship between the density of root filling and the periapical lesion (PAI) at follow-up.

	Periapical Lesion at Follow-Up		
Density of Root Filling	Healthy (Score 1 + 2)	Diseased (Score 3−5)	Total
Adequate	139 (89%)	18 (11%)	157 (100.0%)
Presence of void	27 (77%)	9 (23%)	36 (100.0%)
Total	166 (86%)	27 (14%)	193 (100.0%)

Fisher’s Exact test *p* = 0.099.

**Table 5 dentistry-05-00005-t005:** Relationship between the type of coronal filling and the periapical lesion (PAI) at follow-up.

	Periapical Lesion at Follow-Up		
Type of Coronal Filling	Healthy (Score 1 + 2)	Diseased (Score 3−5)	Total
No filling	4 (50%)	4 (50%)	8 (100.0%)
Composite or Amalgam	51 (91%)	5 (9%)	56 (100.0%)
Crown	103 (85%)	18 (15%)	121 (100.0%)
Temporary restoration	8 (100%)	0 (0%)	8 (100.0%)
Total	166 (86%)	27 (14%)	193 (100.0%)

Fisher’s Exact test *p* = 0.322.

**Table 6 dentistry-05-00005-t006:** Relationship between the quality of the coronal filling and the periapical lesion (PAI) at follow-up.

	Periapical Lesion at Follow-Up		
Quality of Coronal Filling	Healthy (Score 1 + 2)	Diseased (Score 3−5)	Total
No defects	147 (90%)	17 (10%)	164 (100.0%)
Recurrent caries	15 (68%)	7 (32%)	22 (100.0%)
Fracture or Lost/no restoration	4 (57%)	3 (43%)	7 (100.0%)
Total	166 (86%)	27 (14%)	193 (100.0%)

Fisher’s Exact test *p* = 0.001.

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
