# Peer review of "Association of Quality of Coronal Filling with the Outcome of Endodontic Treatment: A Follow-up Study"

_dentistry, 2017, doi:10.3390/dj5010005_

Round 1

Reviewer 1 Report

·         In the first sentence of Materials and Methods: “This is a prospective follow-up study…” I suggest that you delete “prospective”. I agree that this a follow-up study but not a prospective study since the study was not planned and did not start when the root-fillings were performed at baseline.

·         In the title, I suggest that “prospective” is deleted as well.

·         In Table 1, pairwise data are presented and there are frequencies <5 in some fields of the table. Consequently, chi square test is not an appropriate test, I suggest that you use a statistical test for pairwise comparisons like Wilcoxon signed-rank test

·         In Table 2, there are frequencies <5 in some fields of the table. I suggest that you use Wilcoxon rank sum test for these data.

·         What does “N root filling” mean in Table 2?

·         Twenty-three percent of the diseased teeth at follow-up had recurrent caries. I suggest that you discuss the possible mechanism explaining the association between disease in the periapical tissues and presence of caries.   

Author Response

The word “prospective” deleted in the title and the method (Page 2, line 74).

Table 1 (now Table 2) revised based on suggestions from Reviewer-1 and Reviewer-2. Please note that in the revised format of the table, the variables had 2 levels of categorical and matched data, which allowed it for using McNemar’s test. We added one sentence under Statistical analysis (page 3, line 123-124).

Corrected the spelling mistake “N root filling” of Table 3 (revised). Some categories were grouped together because some of the frequencies were <5.

Under discussion (page 7, line 237-240), we discussed the possible mechanism of the association between disease in the periapical tissues and presence of caries.

Reviewer 2 Report

The manuscript presents a prospective follow-up study of root-filled teeth with the aim to assess the effect of the quality of the restoration and of the root-filling for the success of the endodontic treatment. The strength with this study is that the quality of the coronal restorations have been determined both clinically and radiographically, something that is rather uncommon. However information in the Material and method section as well as the presentation of the results need to be improved befor publication can be considered.

1. Material and Methods. The length of the observation period is not clear. The authors write that the study was conducted from 2010-2014. Was it all patients treated in 2010 that was recalled 2014 for a 4-year follow-up examination? Maybe the patients were treated also 2011 and 2012 and the follow-up periods varied. Please present the years when the patients were treated and the year when the follow-up examination was performed and present the length of the follow-up period i.e. a 2-4 year follow-up.

2. It seems as if the radiographic examination was performed using periapical radiographs but it is not clearly presented in M and M section. Please write clearly when, how and what type of radiographic examinations that were performed.

3. Results. Information was available for 96 patients with 212 teeth. The ideal situation is to randomly choose one tooth per patient for examination since individual factors can influence the results of the treatment. If you choose to include all the 212 teeth you should present a frequency table: the number of patients with one tooth treated, the number of patients with 2 teeth treated, the number of patients with 3 teeth treated and so on. Such a table makes it possible for the reader to get an idea of the extent of the individuals' influence on the results.

4. It is written that information is available for 212 teeth, 13 teeth were extracted = 199 teeth available at follow-up. But the authors write in the next sentence that 193 teeth have available data. What about the 6 teeth missing? Data can be unavailable for different reasons, please present information for all the 199 teeth left after the tooth extractions.

5. In the next paragraph it is written "Table 1 shows that at base-line examination 91/190 (48%) where either normal......".  Now the total amount of teeth is 190, how about the 3 teeth not included compared to the 193 in the paragraph above? Please present information for all the 199 teeth. If for example the quality of the radiographs were insufficient for interpretation for x teeth, this can be added in a note under the table. In Table 2 the total amount of teeth is 193 but in Table 3 the total amount of teeth is 192. Please present information for all teeth also for those with radiographs that were not interpretable and other reasons for not being included.

6. It is important that the information is presented in a consistent way. For example in the Result section, the second paragraph and the 4th sentence is written "Among 193 with available data, 91 (47%) had no lesions at baseline, while the majority (52%) had apical lesions...." please write 100 (52%) had apical lesions.

7. In the 3rd paragraph and the 4th sentence was written "Among those who were recorded normal at baseline, the success rate was 93% (85/91) at follow-up. In the next sentence "Among those who have had any PAI lesion......"  also the success rate should be the focus (see point 6 above). Write for example Among those who have any PAI lesion the success rate was 80% (79/99). With such a presentation you can compare the success rates between teeth with normal periapical condition at baseline (PAI 1+2) and those with apical periodontitis (PAI 3+4+5). Please check that the figures are appropriate. Out of the 190 teeth presented in Table 1 only 99 teeth have a PAI score of 3-5.

8. In the section 3.2 Relation between PAI at follow-up with length of root filling the most interesting analysis is the other way round - the relation between the length of the root-filling and PAI at follow up. You are advised to recalculate the figures in Table 2 and put the length of the root fillings in the vertical position and the periapical status (PAI 1+2) and (PAI 3+4+5) in the horisontal position. You are further advised to merge some of the groups, i.e. GP extruded and sealer extruded to one group  (root-filling material extruded). As it is now some of the groups are too small for chi square test. Further I do not understand the group N root-filling, does it mean no root filling? If so, maybe it can be merged with the group > 2 mm. With this recommended analysis you can compare the influence of the different length of the root-fillings on the  periapical status, which is the common and accepted way to do it.  

9. In Table 2 you have found 167 teeth to have no lesion at follow up but in Table 1 there are 164 teeth with no lesion (PAI 1+2) at follow up. Why do you miss 3 teeth in Table 1? It must be explained.

10. For all calculations in Table 3, Table 4 and Table 5, see my comment in point 8. The periapical status should be in the horisontal position and the density of the root-fillings, the type of coronal restoration as well as the quality of the coronal filling should be in the vertical position. This is the accepted way to do the statistical analysis and makes it possible for you to assess the influence of the root-filling quality and the type or quality of the restoration on the periapical status. For example it will probably make it possible for you to write in the 2nd paragraph, 2nd sentence - However density was not significantly associated with AP (PAI 3+4+5) at follow up - instead of the present sentence that says that "PAI at follow up and density was not statistically significant", which is a sentence that does not really mean anything.

11. Also in Table 4 and Table 5 some of the groups are too small. You may consider if the IRM group cam be merged with the Temporary crown group to a temporary restoration group.

12. In the chi square analyses were you have found significant differences, you should present between which groups the differences were found (Table 1 and Table 5).

13. Discussion. The 2nd paragraph, change the 2nd sentence to - Obviously, the success rate was much higher among those who did not show any lesions at baseline compared to those that did (93% and 80% respectively).

14. In the 4th paragraph you claim that there are no other studies that use a combination of clinical and radiographic assessment for the type and quality of the restorations. Recently a study was published which you can consider: Dawson, Petersson, Wolf, Åkerman. Periapical Status of Root-filled Teeth Restored with Composite, Amalgam, or Full Crown Restorations; A Cross-sectional Study of a Swedish Adult Population. J Endod 2016; 42: 1326-1333. In this study both a clinical and a radiographic assessment was made.

15. In the discussion you have used the word incidence were you should use the word prevalence. Incidence means new disease cases during a certain period of time (a week, a month, a year). In the referred articles the prevalence is the focus. Please change incidence to prevalence.

Author Response

Material and Methods: The length of the observation (follow-up) period was defined (Page 2, line 78-80).

2. One sentence was added to clarify the radiographic examination (page 3, line 102).

Results: Table 1 (new) shows the frequency distribution of patients with length of follow-up period according to the number of teeth treated (1, 2, 3, etc.) (please see the new Table 1, page 4).

One sentence added to explain the missing data of 6 patients (Page 3, line 134-135).

Data were corrected for missing values.

Data corrected. The sentence revised as suggested to maintain consistency (page 3, section 3.1, line 142-149.

Table 1 (new Table 2) revised as suggested. Sentences revised (line 142-149) as suggested.

The headings of 3.2 and 3.3 changed as suggested. Some of the groups merged to increase the frequencies where the numbers were too small. The tables were constructed changing the format of the columns and the rows, as suggested.

Former Table 2 (now Table 3) data corrected. The total persons with no lesions are 166, and the figure is consistent with other tables (Table 2, 4, 5, and 6).

All the tables revised as suggested. The sentence on PAI at follow-up revised (Page 5, line 171-172).

IRM group and Temporary crown group are combined as suggested (new table 5)

Data of Chi-square tables are explained to show which groups differ from which others (page 5, line162-164; page 6 line 176-180).

Discussion: The sentence revised as suggested (Page 6, line 190-192).

Added the reference as suggested and the sentence revised (Page 6, line 199-200).

The word “incidence” was replaced by “prevalence” throughout the paper.

Round 2

Reviewer 2 Report

The revised manuscript has improved, but there are still some questions and problems that need to be cleared out.

1. It is very good that this study is a follow-up study and according to Table 1 the follow up period ranges between 1 and 11 years. However the information in the Table differs from the information in the text, presenting that 350 patients treated from 2010 to 2014 were recalled in 2014.This will give a maximum of 4 years follow-up period. Maybe at the follow-up examination also teeth treated before 2010 were included in the study. If so it should be clearly written and explained in the text. Table 1 is very good and clarifying and should not be changed.

2. The length of the follow-up period (for example 1-11 years or a mean of 4.8 years) should be added to the Abstract. Use for example the mean value 4.8 years in the Abstract.

3. The last sentence in the Introduction (The purpose of the present study..) should include information on the length of the follow-up period. Write for example ......to assess the prevalence of apical periodontitis in root canal-treated teeth after 1-11 years.....

4. For the presentation of the results it is good if the authors use the common and generally accepted model. Probably my comments in the first review were not clear enough to explain what I meant. That means that for each group at base-line we want to compare the frequency of healthy teeth at follow up. For Table 2 we want to know if the frequency of healthy teeth at follow up is significantly larger for the teeth that were healthy at baseline (86/93; 92,5%) compared to the teeth that were diseased at base-line (80/100; 80%). This could be easily analysed by Chi square test. If you prefer McNemar's test you have to explain why. The last sentence under 3.1 "The changes of PAI between base-line and follow-up....." is not so interesting and can be deleted. Write instead if you find any significant differences in the frequency of healthy teeth (PAI 1+2) at follow up for teeth with PAI 1+2 at base-line compared to teeth with PAI 3+4+5 at base.line.

5. The same model should be used for all your calculations. In Table 3 you should compare the frequency of healthy teeth (PAI 1+2) at follow up for the length of root-filling 1-2 mm at base-line (124/141 ; 88%) with teeth with >2 mm or no root filling at base-line (16/21; 76%) and with teeth with Root filling material extruded at base-line (8/10; 80%) and Root-filling at radiographic apex (18/21; 86%) at base-line. The Total kolumn to the right should give 100% and the Total row in the bottom should give 166 (86%) and 27 (14%). Since two of the groups have less than 5 teeth in the subgroups maybe Fichers Exact test should be used instead of Chi Square test.

6. Accordingly in Table 4 you should compare the frequency of healthy teeth at follow-up for teeth with Adequate density of root fillings at base-line (139/157; 89%) with teeth with Presence of voids (27/36; 75%) and in Table 5 and 6 follow the same model. It is the frequency of healthy teeth at follow-up for the different base-line groups that we want to compare; we compare the treatment outcome for the different groups. You have presented the distribution of the different  base-line groups within the teeth that were healthy (and the teeth that were diseased) at follow-up, which is not so interesting and not used by other researchers.

7. Both in Table 5 and Table 6 you have problem with very small groups. Maybe you can make more combinations, for example   Composite + Amalgam to one group Composite or Amalgam filling. In Table 6 maybe Fracture and Lost/no restoration can form one group.

8. The text related to the Tables should be changed to present the comparison of the frequencies of healthy teeth at follow-up (the success frequencies) for the different base-line groups and the statistical differences that you maybe have found. The text in this manuscript that presents the distribution of the base-line groups within the healthy  and the diseased group of teeth at follow-up, can be removed.

9. In the Discussion section, 2nd paragraph 3rd sentence "Obviously, the success rate was much higher among...........compared to those that did (93% and 80% respectively)." If you did find any significant differences between these groups you should include that information. 

10. Discussion 5th paragraph row 214. The sentence "The quality of endodontic treatment and coronal restorations, classified by both............." needs a reference.

11. Discussion, last paragraph, row 236. The sentence "For example, 26% of the diseased teeth at follow-up had recurrent caries." can be changed to  >32% of the teeth with recurrent caries were diseased at follow-up<.   This is probably significantly more than teeth with no defects, if so, write that. The following sentences are not understandable. "Disease in the periapical tissues may lead to a loss of connective tissue and to eventual tooth decay. The pathological changes.............. There is a strong consensus that caries or defective restorations in root-filled teeth allow leakage of oral microorganisms  through the root canal to the periapical area were it causes inflammation. Most root-fillings are not able to prevent a long lasting microbial load. Please read textbooks and scientific articles for understanding and references. For example the reference Dawson et al that you included but also many others. 

Author Response

December 12, 2016 

Editor

Dentistry Journal 

Sub: Second revision of manuscript, dentistry 158290 

Dear Editor,

Please find here a copy of our revised manuscript (second revision) based on the reviewers’ comments and suggestions. The following changes were made:

Reviewer 2

The follow-up period was from 2003 to 2014. This is revised in the abstract and on page 2, line 76.

The length of the follow-up period was added in the abstract (line 22-23)

The last sentence of the introduction includes the length of follow-up period (line72).

Used Chi-square test for table 2. Results summarized as suggested.

All the tables reformatted as suggested. Fisher’s Exact test was used if 25% or more of the expected values were less than 5 in any of the tables. This was the case for Table 3, 4, 5, and 6. Data were summarized as suggested.

Data summarized as suggested.

For Table 5, the group “Composite” and “Amalgam” were combined. For Table 6, the two groups “Fracture” and “Lost/no restoration” were combined.

The text revised as suggested.

In Discussion, the p-value added (line 201).

Reference added (line 225).

The last paragraph of Discussion revised as suggested.

We appreciate the reviewers for very useful suggestions which must have improved the quality of the paper.  

Thanks for your kind consideration. 

Sincerely, 

Manal J. Maslamani

Department of Restorative Sciences

Faculty of Dentistry

Kuwait University

P.O. Box: 24923 - Safat 13110

Kuwait

Tel.  +965 94469394

E-mail [email protected]

Round 3

Reviewer 2 Report

This version of the manuscript has improved significantly, however I still have some comments. Since I suppose that it sometimes is hard for the authors to understand my comments I now have made comments and changes direct in the manuscript that is attached to this message.

Author Response

Addressed all the comments and suggestions made by the reviewer in the revised manuscript.

Round 4

Reviewer 2 Report

According to my opinion the manuscript can now be published after one more correction.

Reference 32 does not support the findings described in the end in the discussion. Reference 32 (Igari et al) is all about periodontal disease in relation to general health. It can not support the statement that defect restorations allow leakage of microorganisms between the root-filling and the root-canal to the periapical area were it causes inflammation. Please delete reference 32. The new reference Kirkevang et al (33) supports the statement and can be used.

In my opinion it is a pity that the authors have chosen not to highlight their most interesting finding: the quality of the restoration is significantly associated with the frequency of apical periodontitis at follow-up but the type of restoration is not. There are significant differences in the frequency of healthy teeth (PAI 1+2) at follow up for teeth with adequate restorations compared to non adequate restorations or teeth with caries (adequate restoration give higher frequency of healthy teeth at follow-up).  However there are no significant differences in the frequency of healthy teeth at follow-up for the different type of restoration. This is an interesting and important finding especially since the authors have both clinical and radiographic examinations and a long follow up period (1-11 years).  

Author Response

We appreciate the reviewer for the suggestions. The following changes were made in the revised manuscript:

Reference 32 deleted.

To emphasize on the most important finding, we replaced the first paragraph of Discussion by the findings suggested by the reviewer (Page 6, lines 182-188).